# Study of the Relationship between Mucosal Immunity and Commensal Microbiota: A Bibliometric Analysis

**DOI:** 10.3390/nu15102398

**Published:** 2023-05-20

**Authors:** Shiqi Wang, Jialin Wu, Duo Ran, Guosen Ou, Yaokang Chen, Huachong Xu, Li Deng, Xiaoyin Chen

**Affiliations:** School of Traditional Chinese Medicine, Jinan University, Guangzhou 510632, China; sikeiwong@stu2018.jnu.edu.cn (S.W.); wujialin@stu2020.jnu.edu.cn (J.W.); ramdor@stu2020.jnu.edu.cn (D.R.); ouguosen@stu2022.jnu.edu.cn (G.O.); yiuhong@stu2022.jnu.edu.cn (Y.C.); xuhuachong@jnu.edu.cn (H.X.)

**Keywords:** immunity, microbiota, microecology, mucosa dysfunction, bibliometric analysis

## Abstract

This study presents the first bibliometric evaluation and systematic analysis of publications related to mucosal immunity and commensal microbiota over the last two decades and summarizes the contribution of countries, institutions, and scholars in the study of this field. A total of 1423 articles related to mucosal immunity and commensal microbiota in vivo published in 532 journals by 7774 authors from 1771 institutions in 74 countries/regions were analyzed. The interaction between commensal microbiota in vivo and mucosal immunity is essential in regulating the immune response of the body, maintaining communication between different kinds of commensal microbiota and the host, and so on. Several hot spots in this field have been found to have received extensive attention in recent years, especially the effects of metabolites of key strains on mucosal immunity, the physiopathological phenomena of commensal microbiota in various sites including the intestine, and the relationship between COVID-19, mucosal immunity and microbiota. We hope that the full picture of the last 20 years in this research area provided in this study will serve to deliver necessary cutting-edge information to relevant researchers.

## 1. Introduction

A vast number of commensal microbiota, including bacteria, fungi, viruses, and protozoa, are parasitic in the human body, and it is estimated that human commensal microbes contain approximately 100 trillion cells [1]. They are involved in the regulation of multiple physiological functions in the local and even distant organs of the host organ, modulating the host immune system and releasing metabolites [2,3,4,5]. In addition to the well-known intestinal microbiota, more and more attention has been paid in recent years to commensal microbiota regulation of host physiological functions in the respiratory tract, oral cavity, and vagina.

Undoubtedly, commensal microbiota are exotic substances to the host. After recognition by the host and screening of its immune system (clearance and tolerance), commensal microbiota of a specific structure can be permitted to survive in the body. Figure 1 shows the common mucosal tissue in the human body (Figure 1). Mucosal immunity, the largest component of the entire immune system, is the structure in which the host comes into direct contact with the commensal microbiota. Consisting of a tightly connected surface of mucosal epithelial cells, mucus, and antimicrobial peptides secreted from the mucosal surface and immune cells residing in the lamina propria of the mucosa, the intestinal mucosal barrier prevents the multiplication of pathogens and the invasion of antigenic substances produced by the commensal microbiota [6,7].

The interaction between commensal microbiota and mucosal immunity works in both directions. On the one hand, commensal microbiota and their metabolites impact local mucosal immunity. The supplementation and increase of probiotics facilitates the maintenance of integrity on the mucosal immune barrier against various pathogenic factors [8]. In contrast, conditionally pathogenic bacteria and their products, such as LPS, could activate mucosal immune responses and cause barrier damage [9]. On the other hand, the altered immune status of the organism regulates the composition and metabolism of the commensal microbiota via mucosal immunity. Due to the interconnectedness of mucosal immunity [10], local mucosal immune activation caused by a certain disease tends to induce similar changes in distal mucosal immunity, which in turn influences the composition of the commensal microbiota in distal organs. For example, in the influenza model mice, although no influenza virus could be found in the intestine, there was a disturbance in the intestinal flora, which was associated with simultaneous impairment of the intestinal and pulmonary barriers and reduction of short-chain fatty acid [11]. To be exact, the direct communicative actions of the body’s mucosa modulate the commensal microbiota and also exert systemic immunomodulatory effects through signaling pathways. It is widely known that the immune balance of the body could be modified with oral administration or intranasal inhalation of probiotics or prebiotics, or fecal transplantation of intestinal flora, for the treatment of various immune-related diseases throughout the body, such as inflammatory bowel disease [12], asthma [13], chronic obstructive pulmonary disease [14,15], influenza [16], COVID-19 [17], hepatitis B [18], tuberculosis [19], neuroinflammation-related depression, anxiety disorders [20], etc. Yet, the therapeutic use of prebiotics is limited because they are living organisms. Live probiotics, or non-live microorganisms made from probiotics, as well as the specific route of administration, carry a greater impact on the effectiveness of disease treatment and require more clinical evaluation before they can be widely applied [21]. Overall, mucosal immunity is a key target for commensal microbiota to perform a wide range of immunomodulatory functions, and in-depth studies on it will contribute to revealing the essential role of commensal microbiota in human diseases.

In recent years, with the widespread utilization of microbial 16S rRNA sequencing technology, more and more diseases are considered to be associated with commensal microbiota. We noted that the number of studies in this area has been increasing gradually but there is a lack of bibliometric analysis of studies related to commensal microbiota in vivo and mucosal immunity. It is not conducive for researchers to keep abreast of research evolution trends, as well as current research hotspots, especially when dealing with the topic pertaining to multisystem diseases. Bibliometric analysis refers to the comprehensive analysis of the literature in a field with specific visual analysis software (e.g., Citespace and VOSviewer) and the visualization of information about research collaboration networks, knowledge base networks, disciplinary frontier hotspots, and future research trends [22]. The approach facilitates researchers to track relevant experts, as well as fundamental and predictive key literature [23]. Our study employed both software to visualize and analyze the literature over the past 20 years of academic papers in fields related to commensal microbiota in vivo and mucosal immunity. Based on the results of our analysis, we summarize the current state of research in this area and make predictions for future research trends.

## 2. Method

### 2.1. Data Collection

All literature on commensal microbiota and mucosal immunity was searched for in the core collection of Web of Science. The search results were downloaded as plain text with title, author, abstract, and citation information and were named “download_*”. The text was then saved in the “inputs” folder. Additional information including the author’s h-index and the impact factor of the journal was obtained by searching Web of Science.

### 2.2. Research Strategy

The literature search was completed on 14 February 2023. The search strategy was as follows (TS = (“Mucosal immunity” OR “mucosal immunology” OR “Mucosal Immunization” OR “Mucosal Immune Response” OR “Mucosal Immune Responses”) AND TS = (“microbio*” OR (“flora” AND (“in vivo” OR “gut“ OR “intestinal” OR “gastrointestinal” OR “lung” OR “respiratory” OR “Pulmonary” OR “mucosal” OR “vaginal”)))) AND DT = (Article OR Review) AND LA = (English). The timespan was limited to the period from 1 January 2003 to 31 December 2022.

### 2.3. Data Analysis and Visualization

All tables and trend line graphs of the number of publications in the field were produced by Microsoft 2022. Author and institutional collaboration networks and keyword visualization network graphs were completed by VOSviewer (1.6.18). Visual maps of the global volume of published literature were produced using Scimago Graphica. Citespace is good at identifying current research hotspots in a research field and providing a basis for predicting future research trends since it analyzes keywords and references with high burst values [24]. Therefore, citation timeline graphs as well as keyword and citation burst values were analyzed by Citespace (6.1.R2). The flow chart below demonstrates the whole process of the study (Figure 2).

## 3. Results

### 3.1. Basic Information about the Publication

Searching the Web of Science database and removing one duplicate, we included a total of 1423 papers on mucosal immunity and commensal microbiota research from 2003 to 2022 for this survey. Figure 3 illustrates that the number of papers on mucosal immunity and commensal microbiota has steadily increased over the past two decades, with an overall upward trend. The trend prediction model for the number of papers (R^2^ = 0.9605) indicates that current research on the relationship between commensal microbiota and mucosal immunity is becoming increasingly popular.

### 3.2. Distribution and Cooperation between Countries/Regions and Institutions

Table 1 shows the top 10 countries/regions and the top 10 institutions in terms of number of publications and corresponding percentages. Seventy-four countries/regions have worked on the topic of mucosal immunity and commensal microbiota. Among them, the United States has the highest number of publications (n = 523) with 36.8%. The second highest number of publications is from China (n = 282) with 19.8%. On the institutional side, approximately 1771 institutions contributed to the study of this topic. Within the top 10 most productive institutions, Harvard Medical School ranks first (n = 24), followed by Harvard University (n = 22). The darker the red color shown in the heat map, the higher the number of publications. The U.S. has six of the most productive institutions on the list, while China has three, and Japan has one (Table 1, Figure 4A,B).

Excellent research results certainly cannot be achieved without the collaborative efforts of multiple platforms and resources. In the network diagram of institutional collaboration analysis, the size of the circles indicates the number of publications, and the same color represents the same cluster. In the time-dependent network diagram of institutional collaboration, the color indicates the average start year of each institution’s publications in this research direction (Figure 4C). As shown in Figure 4D, researchers from Harvard University and Institute National de la Recherche Agronomique in France started earlier in the field of mucosal immunity and commensal microbiota research. In contrast, researchers from Nanjing Agricultural University and Huazhong Agricultural University in China have conducted more recent studies.

### 3.3. Conditions about the Author and Co-Cited Authors

Statistically, 7774 authors contributed to articles related to mucosal immunity and commensal microbiota and more than 200 authors have published 3 articles. A total of 732 studies were found to be supported by national government or state health agencies. Hoseinifar, Seyed Hossein (h-index = 44) and Xu, Zhen (h-index = 16) were the most prolific with 13 articles, respectively, followed by Salinas, Irene (h-index = 30), Tlaskalova-Hogenova, Helena (h-index = 31), and Ding, Li-guo (h-index = 8). The tenth-ranked author Shanahan, Fergus has the highest centrality with h-index = 100 (Table 2). Figure 5 shows the collaboration mapping among researchers. The seventeen authors revealed are divided into five colors, representing five clusters among them. Each cluster concentrates on several nearby authors, and frequent synergistic collaborations are usually observed in the same cluster, e.g., Garssen, Johan and Scher, Jose U. While the connections between different clusters are relatively loose, some degree of collaboration also exists between two connected cluster nodes, e.g., Scher, Jose U. and Demoruelle, M. Kristen. It implies that collaboration between research teams/labs conducting research related to mucosal immunity and commensal microbiota is not yet mature.

Co-cited authors are the same authors who are cited in different articles. As shown in Table 3, Macpherson, Andrew J. (n = 292); Hooper, Lora V. (n = 260); and Atarashi, Koji (n = 254) had the maximum number of co-citations. Surprisingly, there was no overlap between the top 10 authors with the most publications and the top 10 authors with the most co-citations, indicating that teams with relatively authoritative research strength and influence on mucosal immunity and commensal microbiota should cooperate deeply, which may create a new breakthrough.

### 3.4. Status of Journal Publication

From 2003 to 2022, 532 academic journals published articles on mucosal immunity and commensal microbiota investigations, 53 of which contained at least 5 articles. The top 15 journals published a total of 378 papers, accounting for 26.56% of all published papers. Ten of these journals had impact factors above 5 in 2022 (Table 4). *Frontiers in Immunology* had the greatest number of publications (n = 98), followed by *Frontiers in Microbiology* (n = 34) and *PLoS ONE* (n = 33).

### 3.5. Co-Citation for References and References with Citation Burst

Two or more articles that are concurrently cited by one or more other later papers are referred to as a co-citation relationship. Thus, articles appearing frequently together in the references are more densely linked. It indicates that the more similar the two are in content, the more closely they are linked. On this basis, the co-cited emergent references are those that have been co-cited with high frequency over a period. Table 5 lists the top 15 co-cited articles ranked by frequency [25,26,27,28,29,30,31,32,33,34,35,36,37,38,39]. Figure 6A illustrates the top 25 citation bursts, with 15 studies cited highly for a sustained period of five years. The article *Induction of intestinal Th17 cells by segmented filamentous bacteria* [19], which has the highest co-citation frequency and the highest citation burst (Strength = 20.38), was published in the journal *Cell* in 2009. It reports that microbial symbiotic regulatory pathways may offer new opportunities for enhancing mucosal immunity and treating autoimmune diseases, providing strong evidence and a solid foundation for the study of mucosal immunity and commensal microbiota. In addition, the following articles still maintain high burst values to date, including *From Dietary Fiber to Host Physiology: Short-Chain Fatty Acids as Key Bacterial Metabolites* (2016) [40], *Gut microbiota, metabolites and host immunity* [41], *DADA2: High-resolution sample inference from Illumina amplicon data* [42], *Reproducible, interactive, scalable and extensible microbiome data science using QIIME 2* [43].

These citations serve as a preliminary basis for further research and an important guide for subsequent studies. Combining the timeline graph and cluster analysis graph, the most frequent words in each cluster were tagged and enriched to discover the research areas. Among them, the top nine clusters in size, including #0 inflammatory bowel disease, #1 innate immunity, #2 dendritic cells, #3 butyrate, #4 autophagy, #5 COVID-19, #6 aging, #7 intestinal flora, #8 epinephelus coioides. In contrast, “#5 COVID-19” is an area of great interest to current researchers as well as an epidemic disease that needs to be addressed globally. In addition, #6 aging is also at the forefront (Figure 6B,C).

### 3.6. Summary from Relevant Research Keywords

Keywords are highly crystallized words that express the topical concepts and summarize the core elements of a publication. In this analysis, a total of 2857 keywords were extracted, of which 124 keywords appeared more than 7 times. In terms of frequency, “mucosal immunity” was the most common keyword (n = 440), followed by “gut microbiota” (n = 182) and “inflammatory bowel disease“ (n = 100), respectively. It indicates that their corresponding fields are valued in relevant studies (Table 6). Figure 7A shows the network visualization of these keywords. The size of the nodes reflects the frequency of the keywords, while the distance between two nodes reflects the strength of their association. Closer keywords are grouped into the same cluster, roughly reflecting the main directions in the field of mucosal immunity and commensal microbiota research. These keywords are clustered by research direction and roughly divided into five categories: the green cluster is related to the intestinal microbiota (commensal flora, Bifidobacterium, etc.) and to mucosal immunity (immunoglobulin A, aryl hydrocarbon receptor, etc.); blue clusters are correlated with diseases of abnormal microbiota (inflammatory bowel disease, Crohn’s disease, bacterial vaginosis, etc.); the red cluster is associated with autoimmunity (food allergy, regulatory T cells, asthma, etc.); the purple cluster is linked to immunity studies in fish (aquaculture, teleost, etc.); and yellow clusters are related to metabolism (short-chain fatty acids, type I diabetes, human milk oligosaccharides, etc.). The graph presents a temporal overlay visualization of the keywords. Earlier appearing keywords are shown in blue, while red indicates the most recent ones. Keywords such as “commensal bacteria”, “toll-like receptors”, and “bacterial vaginosis” are the early main topics. In contrast, the keywords “COVID-19”, “gut-lung axis”, and “short-chain fatty acid” are the most popular topics in recent years (Figure 7B).

Keyword bursts are those keywords cited significantly more frequently over a period of time. Figure 7 shows the top 25 keywords with the strongest citation bursts. “Crohn’s disease” experienced the strongest burst (intensity = 6.83), followed by “short-chain fatty acid” (intensity = 4.56) and “toll-like receptors” (intensity = 4.53), respectively. The keywords “germ-free animals”, “commensal bacteria”, “hygiene hypothesis”, and “Crohn’s disease” received earlier and longer attention during the study period. In addition, “intestinal mucosal immunity”, “vaginal microbiota”, “immune system”, “IgA nephropathy”, “short-chain fatty acids”, “immune response“, “intestinal morphology”, and “gut health” are keywords remaining in an explosive state in 2022. It indicates that these keywords have recently attracted enough attention and may become a hot spot for future research (Figure 7C).

## 4. Discussion

Drawing on data from the Web of Science database from 2003 to 2022, the bibliometric analysis study analyzed the development of research related to mucosal immunity and commensal microbiota in vivo over the past 20 years. There are 1423 articles related to mucosal immunity and commensal microbiota in vivo published by 7774 authors from 1771 institutions in 74 countries/regions in 532 academic journals.

### 4.1. General Condition

During these two decades, articles on mucosal immunity and commensal microbiota have shown a steady increase. Over ten times as many articles were delivered in 2022 as in 2003 (Figure 3), illustrating the growing interest and research exploration in this field. The potential reason for the expansion of research may be due to the growing recognition of the role that commensal microbiota plays in the mechanisms of mucosal immunity [54], thus increasing research funding and effort. It is expected that future global research in this area will continue to increase further in response to mucosal immune system diseases.

The United States is the most localized and funded country for mucosal immunity and commensal microbiota, publishing 523 papers in the last 20 years (Figure 2). Most of the institutions with the ten highest numbers of publications are universities and research institutes, mainly in the United States and China. Among them, Harvard Medical School in Boston, USA, is the institution for the most published papers, with several studies demonstrating that controlling gut microbes in various ways can reduce inflammatory bowel disease [55], ulcerative colitis [56], allergic diseases [57], and tumors [58] along with mucosal immune damage. A collaboration between Harvard Medical School and Nanjing Agricultural University revealed that Ufl1 and Ufbp1, two key components of the Ufm1 E3 ligase, are both highly expressed in intestinal exocrine cells and serve key roles in maintaining intestinal homeostasis and preventing inflammatory diseases [59]. The University of Pennsylvania, together with the Chinese Academy of Sciences, has made new advances in the study of mucosal immune function in fish, demonstrating that the scleractinian skin mucosa shows the same key features as that of mammalian skin, alongside mucosa-associated lymphoid tissue and the diverse microbiota [60]. However, inter-institutional collaboration is limited and homogeneous, which is not conducive to multicenter and multi-targeted in-depth research. Collaboration between countries and institutions in different regions should be further strengthened to promote the development of this field.

Notably, Prof. Seyed Hossein Hoseinifar from Gorgan University of Agricultural Sciences and Natural Resources, Iran, has published the most articles with the highest centrality, mainly focusing on mucosal immunity, microbiomes, and fish diet. In January 2023, his team published an experimental article in which they found that 3% nutmeg powder could be an effective immunostimulant in zebrafish and improve antioxidant defense and stress tolerance [61]. The most cited co-author, Professor Andrew J. Macpherson, is from the University of Bern, Switzerland, whose high-level publications have explored the link between the microbiome and a wide range of immune diseases [62,63]. However, most author-to-author collaborations are limited to intra-team collaborations and few international collaborations. Future collaborations and support can be sought from high-impact, high-centered authors or teams based on the analysis of this study. 

While the journals published in the database are relatively concentrated, analysis of the characteristics of international peer-reviewed journals can help us understand current research directions and hot spots (Figure 4). Among the top 10 most active journals in the field of mucosal immunity and commensal microbiota research, most publishers are located in the United States and Switzerland. Furthermore, the research in the field is multidisciplinary and integrative in nature and covers many aspects, including immunology, microbiology, neurology, molecular biology, medicine, clinical medicine, hygiene, genetics, environmental science, toxicology, nutrition, and more [64]. Undoubtedly, *Frontiers in Immunology*, *Frontiers in Microbiology,* and *PLoS ONE* have published the most relevant articles and remain the most popular journals, which researchers should follow to keep abreast of the latest research trends.

### 4.2. Basic Knowledge Structure

The references that were co-cited more often constitute the basic knowledge of this research area, and these knowledge carriers are presented in Table 5. The top 15 co-cited references consist of 12 experimental articles, 2 reviews, and 1 letter [25,26,27,28,29,30,31,32,33,34,35,36,37,38,39]. Among them, six articles experimentally demonstrate that gut microbes regulate mucosal immunity by affecting short-chain fatty acids or immunoglobulins, and eight articles choose 16S sequencing or macrogenome sequencing to find genetic differences in gut microbes under different conditions. Two reviews address the pivotal role of the gut microbiome in the immune system, immune response, and inflammatory mechanisms. Therefore, short-chain fatty acids (SCFAs) are key mediators for gut microbiota to exert mucosal immune regulation, while Immunoglobulin A (IgA) is a highly concerned effector molecule related to mucosal immunity. In addition, in terms of research methods, combining microbial sequencing with metabolomics analysis of changes in microbial metabolites (especially SCFAs) is the most commonly used research method.

### 4.3. Current Hotspots Analysis and Field Development Prediction

The keywords and references that still maintain high burst values at present would indicate a high level of interest from the past to present time, meaning that they are hotspots for current research. The research directions associated with these keywords and references have a high probability of continuing to be of interest to researchers in this field in the future, and future research directions can be predicted based on these keywords and references. For co-citation references burst value analysis, four references are still blasting away, mainly revolving around key metabolite of bacteria and the effect of pathogenic bacteria on mucosal immunity, as well as presenting the latest tools for microbiome data analysis, which deserve to be explored thoroughly [40,41,42,43] (Figure 6A). Jointly with the keywords that still maintain high burst values (Figure 7C), we make the following analysis of the current research hotspots and future research trends.

#### 4.3.1. Intestinal Microbiota may Regulate Mucosal Immunity through Short-Chain Fatty Acids and Immunoglobulin A

From various studies so far, intestinal microbiota is closely affiliated with mucosal immunity, so what are the pathways through which intestinal microbiota affects the body’s mucosal immunity? One of the microbiota metabolites, short-chain fatty acids, and mucosa-associated immunoglobulin A are under considerable scrutiny (Figure 8).

SCFAs are microbial metabolites that are produced by bacterial fermentation in the gut and exert a variety of effects on the body’s metabolism and immune system [65]. SCFAs have been extensively investigated to determine their utility in sustaining immune homeostasis by regulating mucosal integrity as well as innate and adaptive immunity [66]. Their potential mechanisms involve three aspects [67]. First, SCFAs are engaged in cellular metabolism. Chun et al. [68] found that short-chain fatty acids act on group 3 innate lymphoid-like cells via free fatty acid receptor 2 (FFAR2), on macrophages via metabolic reprogramming, and on memory CD8+ T cells via FFAR2-dependent and FFAR3-dependent T transfer, potentially affecting host defense cell metabolism. Second, SCFAs inhibit histone deacetylases. Short-chain fatty acids induce the production of mucosal tolerogenic dendritic cells [69], macrophages with antimicrobial activity [70], and peripheral regulatory T cells [32] to modulate the mucosal immune system by inhibiting histone deacetylases (HDACs). Third, SCFAs activate G-protein-coupled receptors, as reflected by the direct immunological effects of SCFAs [71]. Acetate is known to be a ligand for GPR43, and propionate is a ligand for both GPR43 and GPR41. The inflammatory response is activated through signal transduction based on the binding of G protein receptors and short-chain fatty acid ligands. Besides, SCFAs regulate immune responses not only in the intestine but also in other distal mucosal sites, such as the lung and respiratory tract. Dysbiosis of the normal intestinal microbiota is an important factor in the development of asthma and other respiratory diseases.

Mucosa-associated IgA is the most abundant immunoglobulin synthesized by the body and generally exists in dimeric form [72]. Concentrations of IgA are highest in mucous membranes and second only to that of IgG in serum. As the first line of immune defense against invasion by bacterial, viral, and fungal pathogens into the host [73,74], IgA resides mainly on the mucosal surface. Secretory IgA (sIgA), one of the IgA isoforms, is the predominant antibody in mucosal immunity and is highly expressed in gastrointestinal tissues. 

With remarkable anti-microbial activity, sIgA binds and neutralizes microorganisms, enhances bacterial agglutination, and prevents cell adhesion, thus keeping microorganisms from entering mucosal epithelial cells. There are several main functions, including immune rejection, growth inhibition, motility inhibition, regulation of bacterial gene expression and metabolism, neutralization, antigen uptake, and association of mucus with IgA-encapsulated bacteria [75,76]. Additionally, sIgA is also found in abundance in other mucosal sites, such as the female genital tract and respiratory lymph nodes [77,78].

#### 4.3.2. The Influence of Microbiota in Other Parts of the Body on Mucosal Immunity

Compared to gut microbiota, studies involving the respiratory microbiome are indeed limited. Initially, it was the differences in the respiratory microbiomes of healthy nonsmokers and smokers that received attention [79]. The oral microbiota of nonsmokers and smokers varied in species such as *Porphyromonas*, *Neisseria*, and *Dictyostelium*, but not in the lung bacterial populations. The lungs of healthy individuals have traditionally been considered sterile organs. However, along with technological developments and research findings, studies have confirmed that microbial communities in respiratory diseases differ from those in healthy subjects and that the pulmonary microbiome influences not only the susceptibility or etiology of respiratory diseases but also the disease activity of respiratory diseases and the corresponding treatment [80]. As a result, the lung is not sterile and not the same nor synchronized with the changes in the intestinal microbiota [81]. Although the relevance of the respiratory microbiome to mucosal immune diseases has been demonstrated [82], insight into the regulatory mechanisms, including host–microbe interactions and the relationship between pulmonary and intestinal microbes, has not yet been gained. It may possibly provide a theoretical basis for proposing new treatments subsequently.

In recent years, other parts of the microbiota have also started to be noticed. Liping Shen et al. (2022) [83] discussed recent evidence on the composition and distribution of female vaginal microecology during different physiopathological periods such as puberty, menstruation, pregnancy, and menopause, proposing a hormone-driven microbial diversity hypothesis to explain the temporal patterns of vaginal microbial diversity during the female reproductive cycle and menopause. Abnormalities in vaginal microecology may induce different metabolic and immune responses or even be clinical markers of various gynecological diseases [84,85]. Anthony J. et al. (2017) [86] suggest that the surface of the eye is also a mucosal site, as that of the intestine, oral cavity, nasopharynx, and vagina. The study proved the presence of a resident commensal microbiome on the ocular surface, which prevent corneal infections by driving the IL-17 response of mucosal γδ T cells, to identify the cellular mechanisms underlying their effects on ocular immune homeostasis and host defense.

#### 4.3.3. The Relationship between COVID-19 and Mucosal Immunity and Microbiota (Potential Applications in Diseases)

To date, according to the World Health Organization (WHO), SARS-CoV-2 has infected more than 766 million people worldwide, with over 6.9 million deaths (https://covid19.who.int/ (accessed on 19 May 2023)). It has been shown that patients with COVID-19 have significantly different gut and lung microbiome compositions compared to non-COVID-19 individuals [87,88]. Pathogens, immunomodulatory probiotics, and tobacco mosaic virus (TMV) are enriched in the lungs of COVID-19 patients. The microbial diversity of their intestinal microbiota was depressed, and the relative abundance of intestinal microbiota with known immunomodulatory potential, such as *Faecalibacterium prausnitzii*, a producer of butyrate, rectal fungi, and *Bifidobacteria*, was reduced, while the relative abundance of *Pseudomonas aeruginosa* was increased [89]. The increase in beneficial bacteria and decrease in harmful bacteria led to abnormal production and expression of SCFAs and IgA, causing excessive inflammatory storms that resulted in acute lung injury, acute respiratory distress syndrome, and multiple organ failure [90,91]. In addition to this, the thinned intestinal mucus layer and reduced luminal surface area compromise the integrity of the intestinal barrier, potentially contributing to severe gastrointestinal symptoms [92]. It is suggested that SARS-CoV-2 caused severe microecological and pulmonary–intestinal axis dysregulation, which brought about abnormal mucosal immune responses. Based on the pathopathologic lineage of SARS-CoV-2 with commensal microbiota and mucosal immunity, modulation targeting the microbiota and immune cells may ameliorate the mucosal immune response to some extent. Exploring these possibilities is an exciting and crucial task for future research.

### 4.4. Benefits and Limitations

This paper presents the first bibliometric evaluation and systematic analysis of publications related to mucosal immunity and commensal microbiota in the last two decades. Our bibliometric study is thorough, clear, and novel because we adopted a systematic search, quantitative statistics, and multidimensional analysis. However, our study does have several drawbacks. The vast majority of articles are in the Web of Science Core Collection database, which may have a small number of omissions and may ignore results outside the English language, representing only a certain degree of most information.

## 5. Conclusions

All in all, our study summarizes base knowledge structure and current hotspots in mucosal immunity and commensal microbiota throughout the past two decennaries and predicts the development trend of this research field. Compared with other articles, the contribution of this study is evident in its rich graphs and ways to reveal the countries/regions, institutions, active journals, core authors/team laboratories, references, and popular keywords that exert great power in the research of this field. The mechanisms by which different symbiotic ecosystems regulate mucosal immunity and the relationship between mucosal immunity, commensal microbiota, and SARS-CoV-2 are likely to be the focus of further research in the future. Our study provides an important resource and a broad perspective on research trends and frontiers for this purpose.

## Figures and Tables

**Figure 1 nutrients-15-02398-f001:**
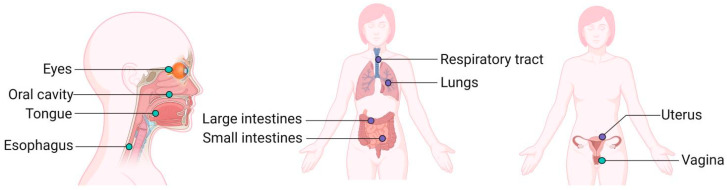
Mucosal Tissues of the Human Body. (Created with BioRender.com, accessed on 17 February 2023).

**Figure 2 nutrients-15-02398-f002:**
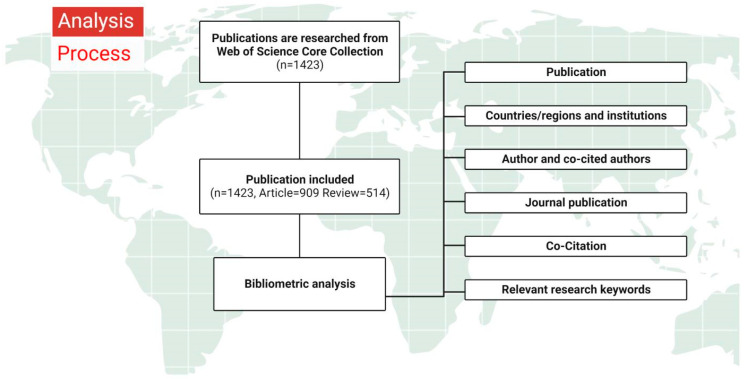
Analysis process. We retrieved 1423 papers on commensal microbiota and mucosal immunity from the core repository of Web of Science. Then, the eligible papers (n = 1423) were screened for inclusion with certain search strategies and time span restrictions. Finally, we analyzed and plotted the data in terms of publication, co-citation, etc.

**Figure 3 nutrients-15-02398-f003:**
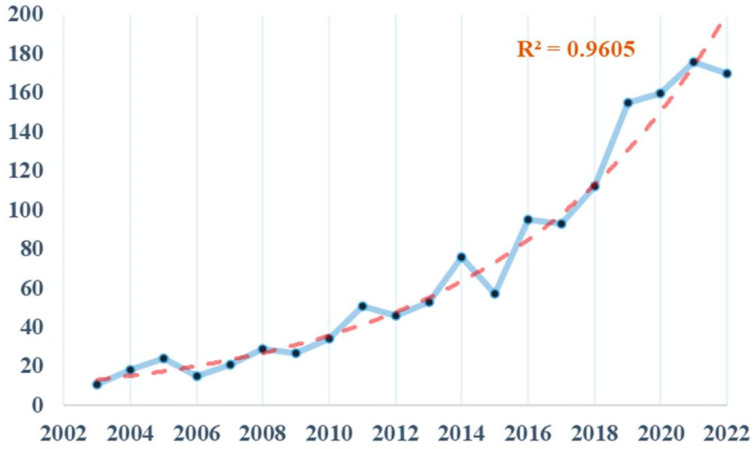
Publications. The number of publications on mucosal immunity and commensal microbiota had steadily increased trend over the past decade.

**Figure 4 nutrients-15-02398-f004:**
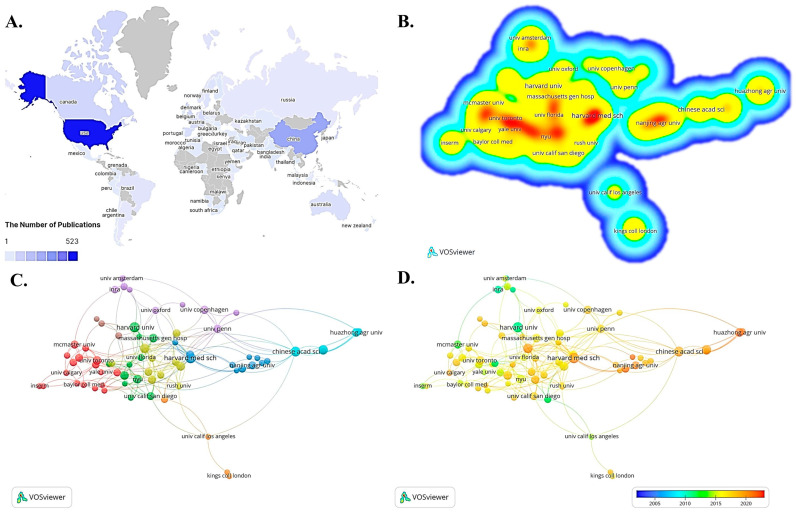
Distribution and cooperation between countries/regions and institutions. (**A**,**B**) Global comparison among countries (**A**) and institutions/organizations (**B**) of the number of research articles on commensal microbiota and mucosal immunity (Published periodical articles ≥ 3). The darker or redder the color, the higher the number of publications, and vice versa. (**C**): Cluster network diagram for the analysis of institutional cooperation in this field (Published periodical articles ≥ 8). There are six colors, and each color represents a cluster. (**D**) Temporal network diagram for the analysis of institutional cooperation in this field (Published periodical articles ≥ 8). Early-stage research institutions are in blue while cutting-edge research institutions are in red.

**Figure 5 nutrients-15-02398-f005:**
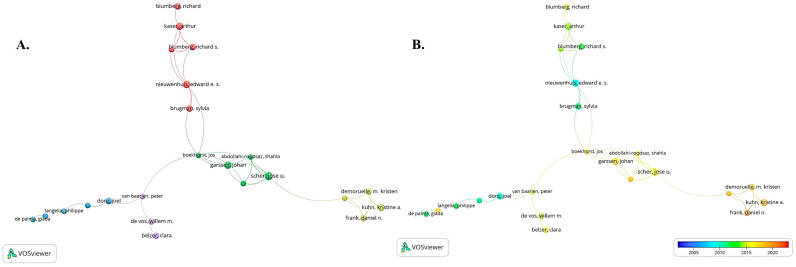
Conditions about the authors. (**A**) Cluster network diagram of the collaborative analysis of authors in this field (Published periodical articles ≥ 3). There are five colors, and each color represents a cluster. (**B**) Time-dependent network diagram of the collaborative analysis of authors in the field (Published periodical articles ≥ 3). Early researchers are shown in blue and frontier researchers in red.

**Figure 6 nutrients-15-02398-f006:**
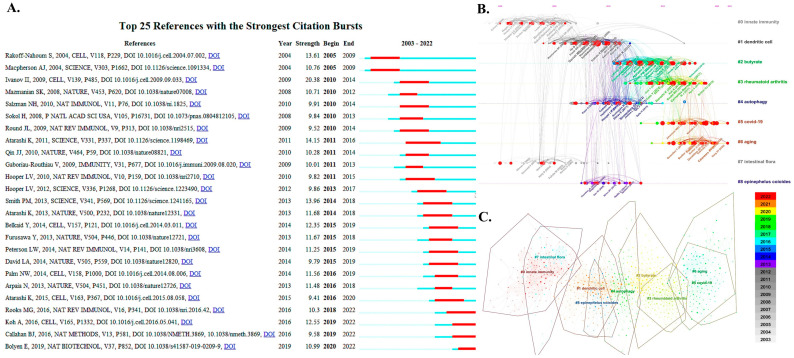
Co-citation for references. (**A**) The top 25 co-citations that burst forth in the field. We mainly focus on the burst intensity as well as the duration of the articles [25,26,27,29,31,32,33,34,35,36,37,38,40,41,42,43,44,45,46,47,48,49,50,51,52,53]. (**B**,**C**) Timeline and cluster analysis graphs of co-citations with high frequency in the field. There are nine frontier areas demonstrated, including #0 inflammatory bowel disease, #1 innate immunity, #2 dendritic cells, #3 butyrate, #4 autophagy, #5 COVID-19, #6 aging, #7 intestinal flora, #8 epinephelus coioides.

**Figure 7 nutrients-15-02398-f007:**
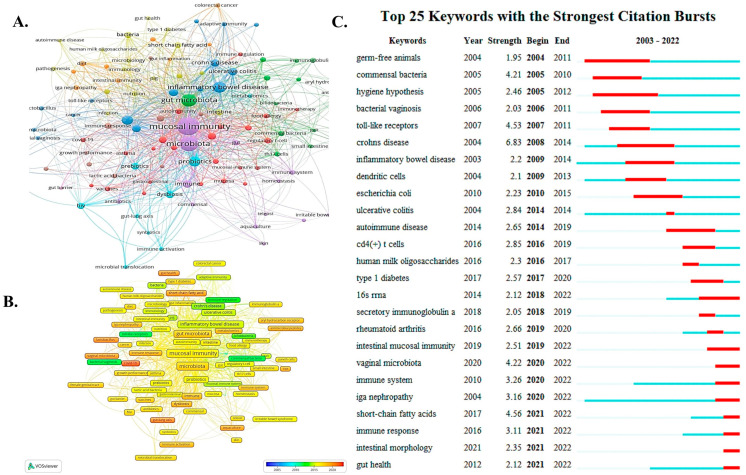
Summary from relevant research keywords. (**A**) Cluster analysis graph of high-frequency keywords in this field (Published periodical articles ≥ 7). Keywords are clustered by research direction and roughly divided into five colors, namely five categories. (**B**) Time correlation analysis graph for high-frequency keywords in this field (Published periodical articles ≥ 7). Early keywords are shown in blue and frontier keywords are shown in red. (**C**) The top 25 keywords that burst forth in the field. We also mainly focus on the burst intensity as well as the duration of the keywords.

**Figure 8 nutrients-15-02398-f008:**
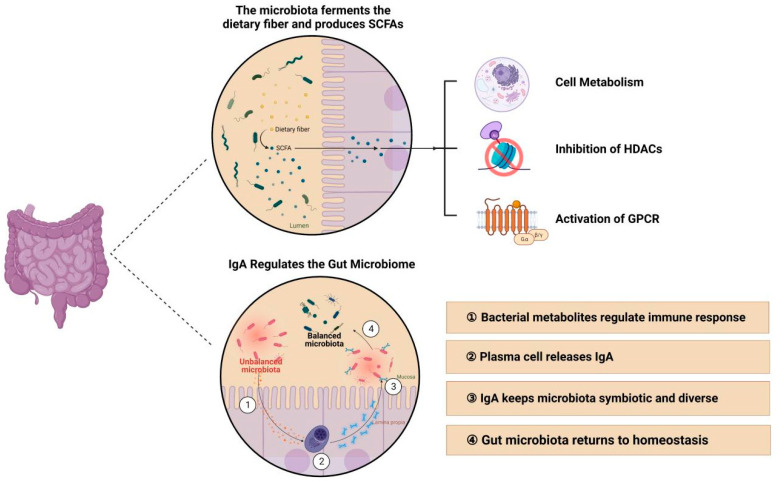
Metabolism of gut microbiota (Created with BioRender.com, accessed on 17 February 2023) [57,65].

**Table 1 nutrients-15-02398-t001:** Top 10 countries and organizations in terms of volume of papers in the field of the relationship between mucosal immunity and commensal microbiota.

Country	Rank	The Number of Publications (% of 1423)	Organization	Rank	The Number of Publications (% of 1423)
USA	1	523 (36.8)	Harvard Med Sch	1	24 (1.7)
China	2	282 (19.8)	Harvard Univ	2	22 (1.5)
England	3	94 (6.6)	Chinese Acad Sci	3	21 (1.5)
Canada	4	89 (6.3)	Univ Washington	4	20 (1.4)
Japan	5	79 (5.6)	China Agr Univ	5	20 (1.4)
Germany	6	78 (5.5)	Univ Toronto	6	17 (1.2)
Italy	7	73 (5.1)	Univ Tokyo	7	17 (1.2)
France	8	70 (4.9)	Univ Calif Davis	8	17 (1.2)
Spain	9	56 (3.9)	NYU	9	17 (1.2)
Iran	10	53 (3.7)	Huazhong Agr Univ	10	17 (1.2)

**Table 2 nutrients-15-02398-t002:** Top 10 authors in terms of paper volume in the field of the relationship between mucosal immunity and commensal microbiota.

Name	Rank	The Number of Publications	H-Index
Xu, Zhen	1	13	16
Hoseinifar, Seyed Hossein	2	13	45
Ding, Li-guo	3	7	8
Tlaskalova-hogenova, Helena	4	7	31
Salinas, Rene	5	7	30
Yu, Yong-yao	6	6	14
Sunyer, J. Oriol	7	6	35
Stepankova, Renata	8	6	30
Scher, Jose U.	9	6	37
Shanahan, Fergus	10	6	100

**Table 3 nutrients-15-02398-t003:** Top 10 co-cited authors of articles in the field of the relationship between mucosal immunity and commensal microbiota.

Name	Rank	The Number of Co-Cited Times	H-Index
Macpherson, Andrew J.	1	292	72
Hooper, Lora V.	2	260	63
Atarashi, Koji	3	254	34
Brandtzaeg, Per	4	243	95
Ivanov, I. I.	5	230	10
Turnbaugh, Peter J.	6	187	52
Round, June	7	172	33
Ley, Ruth E	8	160	75
Johansson, Malin E. V.	9	149	48
Sokol, Harry	10	128	63

**Table 4 nutrients-15-02398-t004:** Top 15 source journals in the field of the relationship between mucosal immunity and commensal microbiota.

Name of the Journal	Rank	The Number of Publications	IF (2021–2022)
*Frontiers in Immunology*	1	98	8.786
*Frontiers in Microbiology*	2	34	6.064
*Plos One*	3	33	3.752
*Mucosal Immunology*	4	25	8.701
*Frontiers in Cellular and Infection Microbiology*	5	22	6.073
*Proceedings of the National Academy of Sciences of the United States of America*	6	21	12.779
*Current Opinion in Gastroenterology*	7	19	2.741
*Fish & Shellfish Immunology*	8	19	4.622
*Immunology*	9	18	7.215
*Microorganisms*	10	16	4.926
*International Journal of Molecular Sciences*	11	16	6.208
*Gut Microbes*	12	15	9.434
*Gastroenterology*	13	15	33.883
*Infection and Immunity*	14	14	3.609
*Nutrients*	15	13	6.706

**Table 5 nutrients-15-02398-t005:** Top 15 co-cited references in the field of the relationship between mucosal immunity and commensal microbiota [25,26,27,28,29,30,31,32,33,34,35,36,37,38,39].

Title	Type	Rank	Year	Cited Times	Journal
Induction of Intestinal Th17 Cells by Segmented Filamentous Bacteria	A	1	2009	133	Cell
Induction of colonic regulatory T cells by indigenous Clostridium species	A	2	2011	93	Science
Recognition of commensal microflora by toll-like receptors is required for intestinal homeostasis	A	3	2004	89	Cell
QIIME allows analysis of high-throughput community sequencing data	L	4	2010	82	Nat Methods
The gut microbiota shapes intestinal immune responses during health and disease	R	5	2009	80	Nat Rev Immunol
Diversity of the human intestinal microbial flora	A	6	2005	78	Science
The microbial metabolites, short-chain fatty acids, regulate colonic Treg cell homeostasis	A	7	2013	76	Science
Metabolites produced by commensal bacteria promote peripheral regulatory T-cell generation	A	8	2013	70	Nature
Molecular-phylogenetic characterization of microbial community imbalances in human inflammatory bowel diseases	A	9	2007	70	Proc Natl Acad Sci USA
Commensal microbe-derived butyrate induces the differentiation of colonic regulatory T cells	A	10	2013	68	Nature
A human gut microbial gene catalogue established by metagenomic sequencing	A	11	2010	68	Nature
Treg induction by a rationally selected mixture of Clostridia strains from the human microbiota	A	12	2013	67	Nature
Induction of protective IgA by intestinal dendritic cells carrying commensal bacteria	A	13	2004	61	Science
Interactions between the microbiota and the immune system	R	14	2012	61	Science
An immunomodulatory molecule of symbiotic bacteria directs maturation of the host immune system	A	15	2004	60	Cell

**Table 6 nutrients-15-02398-t006:** Top 15 keywords in the field of the relationship between mucosal immunity and commensal microbiota.

Rank	Keywords	Counts
1	mucosal immunity	440
2	gut microbiota	182
3	inflammatory bowel disease	100
4	Crohn’s disease	44
5	innate immunity	40
6	barrier function	35
7	ulcerative colitis	34
8	dendritic cells	28
9	short-chain fatty acids	20
10	commensal bacteria	19
11	intestinal epithelium	18
12	B cells	14
13	immune response	14
14	16S rRNA	14
15	gastrointestinal tract	14

## Data Availability

The data used to support the findings are available from the corresponding author upon request.

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
