# Peer review of "Study of the Relationship between Mucosal Immunity and Commensal Microbiota: A Bibliometric Analysis"

_nutrients, 2023, doi:10.3390/nu15102398_

Round 1

Reviewer 1 Report

General Comments: The investigators have undertaken a broad study of emerging publications about the interactions between mucosal immunity and commensal microbiota. The current report provides basic survey data, derived from a novel bibliometric approach and presents interesting and important information about the increase of publications and sources of studies in this area. The analysis and presentation of cocited
references is highly original and informative. Further study will be required to examine and summarize changes in research direction in this area and to examine or predict future research trends.
Specific comments and questions:
1. The term bibliometric analysis usually applies to measuring the influence/ impact of research articles on future research and is used loosely here. Also, this is not a typical systematic investigation based on specific empirical evidence for a pre-specified research question(s). However, the investigators did carry out a substantial investigation, discovered important co-citation relationships and have identified specific themes
co-cited with high frequency.
2. The introduction provides a basic, if oversimplified, summary of mucosal-commensal microbiota interaction, focused on regulation of the respiratory, gastrointestinal and genital tracts, and how some diseases influence local mucosal immune response that can lead to effects on microbiota in remote tissues.
This section would be strengthened if relevant cause and effect relationships were specified and described. Strict adherence to the data in selected referenced studies should be used to guide content, such as for the influenza model (ref 11) where the current text emphasized increase in interferons, in contrast to the published study.
3. The Introduction needs references to support statements about effects of oral treatment with probiotics and prebiotics on asthma and COPD. Limitations of knowledge regarding pro/prebiotics should be identified for some of the diseases mentioned.
4. The Methods section provides information about the research strategy for a single search on one day based on few terms and seems unnecessarily limited.
5. Figure 2: The data show inclusion of review papers. Readers will want to know if authors/ corresponding author also have original (primary source, peer reviewed) papers on the same topic and are qualified experts.
6. Results: Table 1: Since more than one paper may come from the same group of investigators or leading author, the question of how many papers present independent investigations is critical for evaluation.
7. Fig 4 is informative and important and discussed in detail. The data could also be probed at the level of specific research question, type of study, such as experimental or human.
8. Results: Table 1: Readers will want to know about international collaborations on a specific topic and about the composition of authors, such as in collaborations between Harvard Med. Sch. or Harvard U. and China or CAS, where investigators might be authors on papers from both institutions on the same topic and perhaps based on overlapping data bases.
8. Results: The position of the authors is also important- are these individuals working as postdocs in training, fellows, junior faculty, faculty, etc. Is there institutional support, government support, or industrial support for these investigators and these studies?
9. Discussion: The discussion about the results is helpful (4.1-4.3). The analysis of the 358 current research hotspots and future research trends is overly detailed like a textbook (4.3.1-4.3.3).

-

Author Response

Dear reviewer,

Thanks for your opinions, these comments are very helpful to improve the quality of the manuscript. We have carefully revised our manuscript, further clarify the logic of writing for improving the quality of the manuscript. Words in red are the changes I have made in the manuscript. Now I response the comments with a point by point and highlight the changes in revised manuscript. Full details of the files are listed. We sincerely hope that you find our responses and modifications satisfactory with acceptable for publication.

  1. The term bibliometric analysis usually applies to measuring the influence/ impact of research articles on future research and is used loosely here. Also, this is not a typical systematic investigation based on specific empirical evidence for a pre-specified research question(s). However, the investigators did carry out a substantial investigation, discovered important co-citation relationships and have identified specific themes co-cited with high frequency.

We are deeply thankful for you raising the important points. Exactly, it is not a typical systematic survey, but the analysis of existing studies could positively contribute to future research. CiteSpace and vosviewer are citation visualization and analysis softwares focusing on analyzing the potential knowledge embedded in scientific analysis, which are gradually developed in the context of scientometrics and data visualization. Since the structure, pattern and distribution of scientific knowledge are presented through visualization, the graphics visualized by such methods are also called "scientific knowledge maps" [1]. Bibliometric analysis, a comprehensive body of knowledge that integrates mathematics, statistics and bibliography and focuses on quantification [2], refers to the comprehensive analysis of the literature in a field with these specific visual analysis softwares and the visualization of information about research collaboration networks, knowledge base networks, disciplinary frontier hotspots and future research trends [3]. The approach facilitates researchers to track relevant expert, as well as fundamental and predictive key literature [4]. Our study employed both software to visualize and analyze the literature over the past 20 years of academic papers in fields related to commensal microbiota in vivo and mucosal immunity. We hope that the full picture of this research area provided in this study will serve to deliver necessary cutting-edge information to relevant researchers.

  1. The introduction provides a basic, if oversimplified, summary of mucosal-commensal microbiota interaction, focused on regulation of the respiratory, gastrointestinal and genital tracts, and how some diseases influence local mucosal immune response that can lead to effects on microbiota in remote tissues. This section would be strengthened if relevant cause and effect relationships were specified and described. Strict adherence to the data in selected referenced studies should be used to guide content, such as for the influenza model (ref 11) where the current text emphasized increase in interferons, in contrast to the published study.

We are profoundly appreciative to the suggested ideas, which have greatly improved our manuscript. We have edited the details to make it clearer and the guidelines have been strictly followed the data of the selected reference study. The changes to the text are shown below (lines 31-76).

  1. The Introduction needs references to support statements about effects of oral treatment with probiotics and prebiotics on asthma and COPD. Limitations of knowledge regarding pro/prebiotics should be identified for some of the diseases mentioned.

We are deeply thankful for you raising the important points. We have added the appropriate references regarding the effects of probiotic and prebiotic oral therapy on asthma and COPD. At the same time, limitations of probiotic therapy were added, which makes our introduction more comprehensive with a more complete understanding of probiotic therapy. The changes to the text are shown below (lines 68-73).

  1. The Methods section provides information about the research strategy for a single search on one day based on few terms and seems unnecessarily limited.

We are profoundly appreciative to the suggested ideas. We have edited the expression to make it more concise. The changes to the text are shown below (lines 101).

  1. Figure 2: The data show inclusion of review papers. Readers will want to know if authors/ corresponding author also have original (primary source, peer reviewed) papers on the same topic and are qualified experts.

We are deeply thankful for the question. Our group has already conducted rich original research in this research direction with unique views and perspectives. We will continue to deepen the follow-up research and be influential in the field. The following are some of the papers that have been published in our group [5-9].

  1. Results: Table 1: Since more than one paper may come from the same group of investigators or leading author, the question of how many papers present independent investigations is critical for evaluation.

We are profoundly appreciative to the suggested ideas. Searching the Web of Science database and removing one duplicate, we included a total of 1423 papers on mucosal immunity and commensal microbiota research from 2003 to 2022 for this survey. Multiple papers may come from the same group of researchers or lead authors, but duplicate papers have been kicked out. Each independent and original article should raise independently investigated questions that will be used as the main evaluation criteria. However, review also assumes considerable importance, as they summarize prior research while raising new questions to guide the direction of future research, likewise without neglect. In conclusion, I believe that the evaluation criteria should be comprehensive and holistic so as to achieve the purpose for which we conducted this study.

  1. Fig 4 is informative and important and discussed in detail. The data could also be probed at the level of specific research question, type of study, such as experimental or human.

We are deeply thankful for your suggestion, which is very important. As a result, we have identified shortcomings in the current study. However, due to the limitations of the existing data analysis software, it is temporarily not possible to obtain the required data for more levels of analysis. We will make improvements based on your suggestions in the subsequent study protocol and implementation.

  1. Results: Table 1: Readers will want to know about international collaborations on a specific topic and about the composition of authors, such as in collaborations between Harvard Med. Sch. or Harvard U. and China or CAS, where investigators might be authors on papers from both institutions on the same topic and perhaps based on overlapping data bases.

We are profoundly appreciative to the suggested ideas. The different collaborations between multiple institutions were already described in Figure 4 of the original paper. We have also added the details of Inter-institutional cooperation to the discussion to make it more comprehensive. The changes to the text are shown below (lines 323-331).

  1. Results: The position of the authors is also important- are these individuals working as postdocs in training, fellows, junior faculty, faculty, etc. Is there institutional support, government support, or industrial support for these investigators and these studies?

We are deeply thankful for your suggestion, which is very important. Unfortunately, the statistics for the position analysis are not yet possible because the authors' position information is rarely mentioned in the literature. However, a total of 732 studies were found to be supported by national government or state health agencies through re-analysis and screening of the data. We have added the details of founded to make it more comprehensive. The changes to the text are shown below (lines 172-173).

  1. Discussion: The discussion about the results is helpful (4.1-4.3). The analysis of the 358 current research hotspots and future research trends is overly detailed like a textbook (4.3.1-4.3.3).

We are profoundly appreciative to the suggested ideas. 358 current research hotspots and future research trends are the core findings of this study that are worthy of careful analysis and more extensive discussion. We have partially rectified the corresponding contents to make them more concise. The changes to the text are shown below (lines 383-487).

Reference:

[1] Fengjun Xiao, Chengzhi Li, Jiangman Sun, et al. Knowledge Domain and Emerging Trends in Organic Photovoltaic Technology: A Scientometric Review Based on CiteSpace Analysis. Front Chem. 2017 Sep 15;5:67.

[2] Yan Wang, Yuhong Jia, Molin Li, et al. Hotspot and Frontier Analysis of Exercise Training Therapy for Heart Failure Complicated With Depression Based on Web of Science Database and Big Data Analysis. Front Cardiovasc Med. 2021 May 19;8:665993.

[3] Sun G, Dong D, Dong Z, Zhang Q, et al. Drug repositioning: A biblio-metric analysis. Front Pharmacol. 2022;13:974849.

[4] Wang J, Maniruzzaman M. A global bibliometric and visualized analysis of bacteria-mediated cancer therapy. Drug Discov Today. 2022 Oct;27(10):103297.

[5] Huachong Xu, Shiqi Wang, Yawen Jiang, et al. Poria cocos Polysaccharide Ameliorated Antibiotic-Associated Diarrhea in Mice via Regulating the Homeostasis of the Gut Microbiota and Intestinal Mucosal Barrier. Int J Mol Sci. 2023 Jan 11;24(2):1423.

[6] Shengsuo Ma, Bing Yang, Yucong Shi, et al. Adlay (Coix lacryma-jobi L.) Polyphenol Improves Hepatic Glucose and Lipid Homeostasis through Regulating Intestinal Flora via AMPK Pathway. Mol Nutr Food Res. 2022 Dec;66(23):e2200447.

[7] Li Deng, Jiali Yan, Huachong Xu, et al. Prediction of exacerbation frequency of AECOPD based on next-generation sequencing and its relationship with imbalance of lung and gut microbiota: a protocol of a prospective cohort study. BMJ Open. 2021 Sep 2;11(9):e047202.

[8] Li Deng, Yucong Shi, Pei Liu et al. GeGen QinLian decoction alleviate influenza virus infectious pneumonia through intestinal flora. Biomed Pharmacother. 2021 Sep;141:111896.

[9] Li Deng, Huachong Xu, Pei Liu, et al. Prolonged exposure to high humidity and high temperature environment can aggravate influenza virus infection through intestinal flora and Nod/RIP2/NF-κB signaling pathway. Vet Microbiol. 2020 Dec;251:108896.

We tried our best to improve the manuscript and made some changes in the manuscript. These changes will not influence the content and framework of the paper. We appreciate for Editors’ and Reviewers’ warm work earnestly, and hope that the correction will meet with approval.

Once again, thank you very much for your comments and suggestions.

Best Regards.

Yours Sincerely,

Huachong Xu

Reviewer 2 Report

Comments for the authors:

In the present manuscript authors presents the first bibliometric evaluation and systematic analysis of publications related to mucosal immunity and commensal microbiota in the last two decades, and summarizes the contribution of countries, institutions and scholars in the study of this field. The aim of the study is very interesting and important. However, there were few deficiencies in the study and the manuscript. It seems that manuscript is written in a hurry and contains still a lot of carelessness in style and writing. The manuscript needs extensive proof-reading by a native English speaking person.

Major concerns:

-Q1: How reliability of the studies included in this study were evaluated? Did authors exclude potential predator journals? Further, was there a IF-limit which journals were included in the study? These concerns need to be somehow commented in the manuscript.

-Q2: Using the word “flora” in the search is old-fashioned an the word “flora” shouldn´t be used in the studies of microbiota anymore. However, the search was done for the publications from 2003 so it is appropriate word in search of publication but it shouldn´t be used in this publication anymore so please remove all the  “flora” -words and replace them using other expressions such as microbiota etc

-Q3: Are all these studies included in vivo studies? This should be underlined already in abstract. Now it is said at the beginning of the discussion.

-Q4: If researchers are mentioned by name as these are: “Of note, Prof. Hoseinifar, Seyed Hossein from Gorgan University of Agricultural Sci- 313 ences and Natural Resources, Iran, has published the most articles with the highest cen- 314 trality, mainly focusing on mucosal immunity, microbiome and fish diet. In January 2023, 315 his team published an experimental article in which they found that 3% nutmeg powder 316 could be an effective immunostimulant in zebrafish and improve antioxidant defense and 317 stress tolerance [43]. The most cited co-author, Professor Macpherson, Andrew J., is from 318 the University of Bern, Switzerland, whose high-level publications have explored the link 319 between the microbiome and a wide range of immune diseases [44-45]”. The IFs of the journals published the articles should be somehow indicated.

Minor concerns:

-Q5: line 24: “A vast number of commensal microbiota, including bacteria, fungi, viruses, and pro- 24

tozoa, are parasitic in the human body, and it is estimated that human microbes contain 25

approximately 100 trillion cell”. Should this be commensal microbes instead of microbiota?

-Q6: line 32: commensal microbiota is; not are…

-Q7: line 33: “After recognition of the host and screening of the immune system (clearance and tolerance), commensal microbiota of a specific structure can be permitted to survive in the body” Who is recognizing and screening? Microbes or the host?

-Q8: line 35: organism´s

-Q9: lines 35-36: same facts about Mucosal immunity as the largest component of the entire organism's immune system is described twice. Please remove one of the those.

-Q10: line 63: Influenza is not a similar immune-related disease as the rest of the diseases in the list. If influenza is included then other bacterial and/or viral diseases should be included as well or influenza should be removed from the list.

-Q11: Figure 1 is not linked with the text. I don´t see connection between the text and the figure. “Consisting of a tightly connected surface of mucosal epithelial cells, mucus and antimicrobial peptides secreted from the mucosal surface, and immune cells residing in the lamina propria of the mucosa, the intestinal mucosal barrier prevents the multiplication of pathogens and the invasion of antigenic substances produced by the commensal microbiota [6-7]. (Figure 1) “. In the figure 1 the mucosal tissues of the human body are described.

-Q12: line 68: 16S rRNA

-Q13: line 92: The search was done on February 14th of 2023

-Q14: line 99: subtitle should be ”Data analysis and visualization”.

-Q15: Lines 112-113 have nothing to do with the figure “Finally, we analyzed and 112 made kinds of graphs by Microsoft 2022, VOSviewer, Scimago Graphica, and Citespace”

-Q16: Figure 2: What is difference between  publication and publication journal? Describe more precisely.

-Q17: table 2 and 3: Names should be written using capital first letter

-Q18: Is the information in table 2 relevant? Why?

-Q19: Format, fonts, style etc are different in all the tables in the manuscript. It seems that the manuscript is not proof-read and the layout hasn´t be gone through carefully. Same can be seen in style and fonts of subtitles

-Q20: Line 291: in vivo should be written in italics

 -Q21: Line 295: during these two decades

-Q22: lines 296-297: “. Over 10 times as many articles were delivered in 2022 as in 2003 296 (Figure 1)” This line has nothing to do with figure 1.

-Q23: line 302: “The United States is the most researched country”. Is it? Or are the most of research done in US?

-Q24: lines 305-306: “Among them, Harvard Medical School in 305 Boston, USA is the most published institution”. Or should it be that most of the studies published are done in Harvard medical School?

-Q25: line 343: SCFAs. The abbreviation should be opened and clarified what it means. When you have once write it open and used the abbreviation, then the abbreviation should be used in the rest of the manuscript. Now the abbreviation was used before it was explained in line 369. Same with IgA… Don´t use abbreviation before it has been explained what it means.

-Q26: Line 373: their rather than its

-Q27: “Mucosa-associated immunoglobulin A (IgA) is the most abundant immunoglobulin synthesized by the body and generally exists in dimeric form [55]. Concentrations of IgA is highest in mucous membranes and second only to that of IgG in serum. As the first line of immune defense against invasion by bacterial, viral and fungal pathogens into the hostIn terms [56-57], IgA resides mainly on the mucosal surface. Secretory IgA, one of the IgA  isoforms, is the predominant antibody in mucosal immunity and is highly expressed in gastrointestinal tissues”. These sentences are twice in the text.

-Q28: sIgA. This abbreviation wasn´t explained

-Q29: Porphyromonas, Neisseria, and Dictyostelium. Bacterial genus should be written using italics

-Q30: a year should be mentioned “Liping Shen et 425 al. [66]”. Shen et al? Anthony J St. Leger et al. [69]? Year? St. Leger et al?

-Q31: line 449: Pseudomonas aeruginosa. Italics!

-Q32: lines 450-451: short-chain fatty acids and immunoglobulin A. Use abbreviations because you have used them before.

-Q33: line 463: last decade or last two decades?

It seems that manuscript is written in a hurry and contains still a lot of carelessness in style and writing. The manuscript needs extensive proof-reading by a native English speaking person.

Author Response

Dear reviewer,

Thanks for your opinions, these comments are very helpful to improve the quality of the manuscript. We have carefully revised our manuscript, further clarify the logic of writing for improving the quality of the manuscript. Words in red are the changes I have made in the manuscript. Now I response the comments with a point by point and highlight the changes in revised manuscript. Full details of the files are listed. We sincerely hope that you find our responses and modifications satisfactory with acceptable for publication.

Major concerns:

-Q1: How reliability of the studies included in this study were evaluated? Did authors exclude potential predator journals? Further, was there a IF-limit which journals were included in the study? These concerns need to be somehow commented in the manuscript.

We are profoundly appreciative to the suggested ideas. Searching the Web of Science database and removing one duplicate, we included a total of 1423 papers on mucosal immunity and commensal microbiota research from 2003 to 2022 for this survey. Web of ScienceTM contains over 21,800 of the world's most authoritative and high-impact academic journals covering the natural sciences, engineering, biomedical sciences, social sciences, arts and humanities, and more. Based on the Bradford's Law of bibliometrics, the database selects only the most important academic journals in each subject area without bias [1]. According to this premise, we considered publications from the web of science database to be reliable and contain extremely few potential predator journals. In addition, the inclusion criteria did not contain restrictions on impact factors to ensure that a full picture of the study area was demonstrated.

-Q2: Using the word “flora” in the search is old-fashioned an the word “flora” shouldn´t be used in the studies of microbiota anymore. However, the search was done for the publications from 2003 so it is appropriate word in search of publication but it shouldn´t be used in this publication anymore so please remove all the “flora” -words and replace them using other expressions such as microbiota etc

We are sorry for this error and apologize for it. We have changed the expression of flora to microbiota, but retained it in the search of the publication. The changes to the text are shown in full text.

-Q3: Are all these studies included in vivo studies? This should be underlined already in abstract. Now it is said at the beginning of the discussion.

We are deeply thankful for your suggestion, which is very important. All these studies were comprised of in vivo studies. We have edited the expression to make it clearer. The changes to the text are shown below (lines 7-19).

-Q4: If researchers are mentioned by name as these are: “Of note, Prof. Hoseinifar, Seyed Hossein from Gorgan University of Agricultural Sci- 313 ences and Natural Resources, Iran, has published the most articles with the highest cen- 314 trality, mainly focusing on mucosal immunity, microbiome and fish diet. In January 2023, 315 his team published an experimental article in which they found that 3% nutmeg powder 316 could be an effective immunostimulant in zebrafish and improve antioxidant defense and 317 stress tolerance [43]. The most cited co-author, Professor Macpherson, Andrew J., is from 318 the University of Bern, Switzerland, whose high-level publications have explored the link 319 between the microbiome and a wide range of immune diseases [44-45]”. The IFs of the journals published the articles should be somehow indicated.

We are profoundly appreciative to the suggested ideas. Impact Factor (IF) is a metric that measures the number of citations and the number of articles published in a journal, reflecting the quality and influence of its publications. However, the impact factor is not the only evaluation criterion. Some journals may be highly influential in certain fields of academic research, but their impact factor is less [2]. Therefore, when we conduct this study, we need to consider the academic level, field influence and impact factor of journals comprehensively, rather than a single criterion. We apologize that for now, it is not marked with the impact factor for the cited literature within the paper.

Minor concerns:

-Q5: line 24: “A vast number of commensal microbiota, including bacteria, fungi, viruses, and pro- 24 tozoa, are parasitic in the human body, and it is estimated that human microbes contain 25 approximately 100 trillion cell”. Should this be commensal microbes instead of microbiota?

We are profoundly appreciative to the suggested ideas. We have changed the expression, which are shown below (line 24).

-Q6: line 32: commensal microbiota is; not are…

We are sorry for this error and apologize for it. We have changed the expression and the text are shown below (line 31).

-Q7: line 33: “After recognition of the host and screening of the immune system (clearance and tolerance), commensal microbiota of a specific structure can be permitted to survive in the body” Who is recognizing and screening? Microbes or the host?

We are deeply thankful for the suggested ideas. It is the host identifying and screening. We have changed the expression and the text are shown below (line 32).

-Q8: line 35: organism´s

We are sorry for this error and apologize for it. We have removed the duplicates and the text are shown below (line 36).

-Q9: lines 35-36: same facts about Mucosal immunity as the largest component of the entire organism's immune system is described twice. Please remove one of the those.

We are sorry for this error and apologize for it. We have changed the expression and the text are shown below (lines 34-35).

-Q10: line 63: Influenza is not a similar immune-related disease as the rest of the diseases in the list. If influenza is included then other bacterial and/or viral diseases should be included as well or influenza should be removed from the list.

We are profoundly appreciative to the suggested ideas. We have changed the expression to make it more accurate. The changes to the text are shown below (lines 64-70).

-Q11: Figure 1 is not linked with the text. I don´t see connection between the text and the figure. “Consisting of a tightly connected surface of mucosal epithelial cells, mucus and antimicrobial peptides secreted from the mucosal surface, and immune cells residing in the lamina propria of the mucosa, the intestinal mucosal barrier prevents the multiplication of pathogens and the invasion of antigenic substances produced by the commensal microbiota [6-7]. (Figure 1) “. In the figure 1 the mucosal tissues of the human body are described.

We are sorry for this error and apologize for it. Mucosal immunity, the largest component of the entire organism's immune system, is the structure in which the host comes into direct contact with the commensal microbiota. Figure 1 shows the common mucosal tissue in the human body. We have changed the expression and the text are shown below (line 35).

-Q12: line 68: 16S rRNA

We are sorry for this error and apologize for it. We have changed the expression and the text are shown below (line 77).

-Q13: line 92: The search was done on February 14th of 2023

We are sorry for this error and apologize for it. We have changed the expression and the text are shown below (line 101).

-Q14: line 99: subtitle should be ”Data analysis and visualization”.

We are sorry for this error and apologize for it. We have changed the expression and the text are shown below (line 108).

-Q15: Lines 112-113 have nothing to do with the figure “Finally, we analyzed and 112 made kinds of graphs by Microsoft 2022, VOSviewer, Scimago Graphica, and Citespace”

We are sorry for this error and apologize for it. According to the content of Figure 2, we have changed the expression and the text are shown below (lines 121-123).

-Q16: Figure 2: What is difference between publication and publication journal? Describe more precisely.

We are profoundly appreciative to the suggested ideas. What is the difference between a publication and a publication journal? A publication/paper refers to a work that addresses a specialized research topic. Depending on the length of an academic paper, it can be further divided into single academic papers, series of academic papers and academic monographs [3]. Journal is a regular publication, such as weekly, decennial, semi-monthly, monthly, quarterly, semi-annual, annual, etc. It is published by a legally established journal unit, which holds International Standard Serial Number and Journal Publication License [4]. They are completely different concepts.

-Q17: table 2 and 3: Names should be written using capital first letter

We are sorry for this error and apologize for it. We have changed the words and the text are shown below (table 2 and 3).

-Q18: Is the information in table 2 relevant? Why?

We are deeply thankful for the question. The information in Table 2 is relevant. From the perspective of authors, the number of published articles and H-index can be part of the criteria to evaluate the influence and status of authors in the field about mucosal immunity and commensal microbiota research. In this study, Hoseinifar, Seyed Hossein(H-index =44) and Xu, Zhen(H-index =16) were the most prolific with 13 articles respectively, followed by Salinas, Irene (H-index =30), Tlaskalova-hogenova, Helena(H-index =31) and Ding, Li-guo(H-index =8). The tenth ranked author Shanahan, Fergus has the highest centrality with h_index=100 (Table 2). The collaboration between authors is relatively close, with 5 authors in the TOP 10 of publication numbers whose H-index is greater than 30. High impact and centrality authors or teams may be sought for future collaboration and support based on the analysis of this study.

-Q19: Format, fonts, style etc are different in all the tables in the manuscript. It seems that the manuscript is not proof-read and the layout hasn´t be gone through carefully. Same can be seen in style and fonts of subtitles

We are sorry for this error and apologize for it. We have edited the format, font and style, which were shown in all tables and subtitles.

-Q20: Line 291: in vivo should be written in italics

We are sorry for this error and apologize for it. We have changed the words and the text are shown below (line 304).

-Q21: Line 295: during these two decades

We are sorry for this error and apologize for it. We have added the words and the text are shown below (line 308).

-Q22: lines 296-297: “. Over 10 times as many articles were delivered in 2022 as in 2003 296 (Figure 1)” This line has nothing to do with figure 1.

We are sorry for this error and apologize for it. We have changed the figure number and the text are shown below (line 310).

-Q23: line 302: “The United States is the most researched country”. Is it? Or are the most of research done in US?

We are profoundly appreciative to the suggested ideas. We have changed the expression to make it more cautious. The changes to the text are shown below (line 316).

-Q24: lines 305-306: “Among them, Harvard Medical School in 305 Boston, USA is the most published institution”. Or should it be that most of the studies published are done in Harvard medical School?

We are deeply thankful for the suggested ideas. We have changed the expression to make it more cautious. The changes to the text are shown below (lines 320-321).

-Q25: line 343: SCFAs. The abbreviation should be opened and clarified what it means. When you have once write it open and used the abbreviation, then the abbreviation should be used in the rest of the manuscript. Now the abbreviation was used before it was explained in line 369. Same with IgA… Don´t use abbreviation before it has been explained what it means.

We are sorry for this error and apologize for it. We have changed the expression and the text are shown below (lines 365-367).

-Q26: Line 373: their rather than its

We are sorry for this error and apologize for it. We have changed the expression and the text are shown below (line 397).

-Q27: “Mucosa-associated immunoglobulin A (IgA) is the most abundant immunoglobulin synthesized by the body and generally exists in dimeric form [55]. Concentrations of IgA is highest in mucous membranes and second only to that of IgG in serum. As the first line of immune defense against invasion by bacterial, viral and fungal pathogens into the hostIn terms [56-57], IgA resides mainly on the mucosal surface. Secretory IgA, one of the IgA  isoforms, is the predominant antibody in mucosal immunity and is highly expressed in gastrointestinal tissues”. These sentences are twice in the text.

We are sorry for this error and apologize for it. We have removed the duplicates and the text are shown below (lines 419-425).

-Q28: sIgA. This abbreviation wasn´t explained

We are sorry for this error and apologize for it. We have explained it and the text are shown below (line 417).

-Q29: Porphyromonas, Neisseria, and Dictyostelium. Bacterial genus should be written using italics

We are sorry for this error and apologize for it. We have edited the words and the text are shown below (line 438).

-Q30: a year should be mentioned “Liping Shen et 425 al. [66]”. Shen et al? Anthony J St. Leger et al. [69]? Year? St. Leger et al?

We are deeply thankful for the suggested ideas. We have added the year to make it more cautious. The changes to the text are shown below (line 452, 458).

-Q31: line 449: Pseudomonas aeruginosa. Italics!

We are sorry for this error and apologize for it. We have edited the words and the text are shown below (line 475).

-Q32: lines 450-451: short-chain fatty acids and immunoglobulin A. Use abbreviations because you have used them before.

We are profoundly appreciative to the suggested ideas. We have changed the expression and the text are shown below (line 477).

-Q33: line 463: last decade or last two decades?

We are sorry for this error and apologize for it. We have edited the words and the text are shown below (line 490).

Reference:

[1] Michael Gusenbauer, Neal R Haddaway. Which academic search systems are suitable for systematic reviews or meta-analyses? Evaluating retrieval qualities of Google Scholar, PubMed, and 26 other resources. Res Synth Methods. 2020 Mar;11(2):181-217.

[2] Emmanuel J Favaloro. Measuring the quality of journals and journal articles: the impact factor tells but a portion of the story. Semin Thromb Hemost. 2008 Feb;34(1):7-25.

[3] W C Peh, K H Ng. Basic structure and types of scientific papers. Singapore Med J. 2008 Jul;49(7):522-5.

[4] Randy G Floyd, Kathryn M Cooley, James E Arnett. An overview and analysis of journal operations, journal publication patterns, and journal impact in school psychology and related fields. J Sch Psychol. 2011 Dec;49(6):617-47.

We tried our best to improve the manuscript and made some changes in the manuscript. These changes will not influence the content and framework of the paper. We appreciate for Editors’ and Reviewers’ warm work earnestly, and hope that the correction will meet with approval.

Once again, thank you very much for your comments and suggestions.

Best Regards.

Yours Sincerely,

Huachong Xu

Reviewer 3 Report

Very interesting work, remarkable bibliographic entries.

I really liked the paragraphs 4.3.1, 4.3.2 and 4.3.3, well explained.

Tables 1 and 2 should be arranged (e.g. initials of proper names and organizations in capital letters).

Author Response

Dear reviewer,

Thanks for your opinions, these comments are very helpful to improve the quality of the manuscript. We have carefully revised our manuscript, further clarify the logic of writing for improving the quality of the manuscript. Words in red are the changes I have made in themanuscript. Now I response the comments with a point by point and highlight the changes in revised manuscript. Full details of the files are listed. We sincerely hope that you find our responses and modifications satisfactory with acceptable for publication.

Tables 1 and 2 should be arranged (e.g. initials of proper names and organizations in capital letters).

We are profoundly appreciative to the suggested ideas. We have edited the words to make it more precise. The changes to the text are shown below (Tables 1 and 2).

We tried our best to improve the manuscript and made some changes in the manuscript. These changes will not influence the content and framework of the paper. We appreciate for Editors’ and Reviewers’ warm work earnestly, and hope that the correction will meet with approval.

Once again, thank you very much for your comments and suggestions.

Best Regards.

Yours Sincerely,

Huachong Xu
